# Effects of Propofol on Cortical Electroencephalograms in the Operation of Glioma-Related Epilepsy

**DOI:** 10.3390/brainsci13040597

**Published:** 2023-03-31

**Authors:** Xin Li, Yu Wei, Yanfeng Xie, Quanhong Shi, Yan Zhan, Wei Dan, Li Jiang

**Affiliations:** Department of Neurosurgery, The First Affiliated Hospital of Chongqing Medical University, Chongqing 400016, China; lixincocosulate@163.com (X.L.); wyulala2023@163.com (Y.W.); xyf3058@163.com (Y.X.); shqh69@yahoo.com (Q.S.); zhan.yan2@hotmail.com (Y.Z.)

**Keywords:** propofol, epilepsy, cortical electroencephalogram, glioma-related epilepsy, epilepsy surgery

## Abstract

Background: A cortical electroencephalogram (ECoG) is often used for the intraoperative monitoring of epilepsy surgery, and propofol is an important intravenous anesthetic, but its effect on EEGs is unclear. Objectives: To further clarify the effect of propofol on cortical ECoGs during glioma-related epilepsy surgery and to clarify the possible clinical value. Methods: A total of 306 patients with glioma were included in the study. Two hundred thirty-nine with glioma-related epilepsy were included in the epilepsy group, and 67 without glioma-related epilepsy were included in the control group. All patients experienced continuous, real-time ECoG monitoring and long-term follow-up after surgery. Results: After injection of low-dose propofol, the rate of activated ECoGs in the epilepsy group (74%) was significantly higher than in the control group (9%). Furthermore, compared with patients in the untreated group, patients in the treated group had lower rates of early and long-term postoperative seizure frequencies and fewer interictal epileptiform discharges (IEDs). Conclusions: Low-dose infusion of propofol can specifically activate ECoGs in epilepsy patients. Therefore, activated ECoGs might provide an accurate and reliable method for identifying potential epileptic zones during glioma-related epilepsy surgery, resulting in better early and long-term prognoses after epilepsy surgery.

## 1. Introduction

Glioma is considered the most common and deadly brain tumor, accounting for about 81% of malignant tumors in the brain [1]. Symptomatic epilepsy secondary to glioma is glioma-related epilepsy (GRE). About 30% to 90% of glioma patients complain of epilepsy as the first clinical manifestation, usually caused by the mass effect of a tumor and peritumoral edema [2]. Its antiepileptic drugs are ineffective, and surgical resection of the tumor is the primary treatment for most patients.

However, after removing the tumor, 20–30% of patients still cannot achieve postoperative seizure freedom [3,4]. Therefore, the current surgical method of GRE advocates ‘supratotal resection’, that is, resection of tumors and epileptic foci [5]. As a result, in addition to resection of the tumor, identifying the locations of potential epileptic foci accurately is also of vital importance [6].

At present, there are numerous methods used to localize epileptic foci, for example, the electroencephalogram (EEG), magnetic resonance imaging (MRI), and positron emission tomography (PET) [7]. The intraoperative cortical electroencephalogram (ECoG) is an important tool to localize epileptic foci in real time, which provides an important guide to the neurosurgeon, particularly in epilepsy surgery [7,8]. The ECoG is based on interictal epileptiform discharges (IEDs), such as spike sharp waves, sharp slow waves, and multi-spike waves. This is the electrophysiological basis of the localization of epileptic foci.

Since its launch in 1986, propofol has been widely used for general anesthesia induction and maintenance because of its rapid redistribution and short elimination half-life [9]. These properties are essential during neurosurgical operations, especially epilepsy operations [10]. However, the effects of propofol on brain electrical activity are still controversial. On the one hand, it is a potent antiepileptic drug [11]; on the other hand, it may trigger epileptic seizures [12,13].

In this study, to further validate the effects and significance of propofol on ECoG, early and long-term outcomes of patients with GRE were analyzed to test the value of propofol for epilepsy surgery.

## 2. Materials and Methods

### 2.1. Patients

This study was in line with the human research guidelines approved by the Clinical Research Ethics Committee of the First Affiliated Hospital of Chongqing Medical University. Written and informed consent was obtained from all patients or their legal guardians.

Patients admitted to the hospital from January 2013 to December 2021 were included in the study. The patients were divided into epilepsy and control groups (Figure 1).

The inclusion criteria of the epilepsy group were: (1) epilepsy as the primary presentation and with EEGs showing definite epileptic discharge before surgery, (2) the pathological examination results of the operation confirmed neuroglioma, and (3) patients who accepted surgical treatment, intraoperative neuronavigation, and ECoG monitoring during surgery.

The inclusion criteria of the control group were: (1) no seizure presentation or evident IEDs in EEGs before surgery, (2) neuroglioma confirmed by pathological examination results, and (3) patients who accepted surgery, intraoperative neuronavigation, and ECoG monitoring during surgery.

The exclusion criteria of the patients were: (1) no gross total resection (GTR) or near gross total resection (NTR) of glioma, (2) follow-up time <12 months, (3) were diagnosed with recurrent glioma or metastases, and (4) patients with uncompleted clinical data or data lost in the follow-up.

The epilepsy and control groups were further divided into the activated and inactivated groups, depending on whether the ECoG was activated after low-dose propofol infusion. The spike amplitude, spike frequency, and the number of spiking leads of ECoGs were quantified and analyzed for a 2 min visual counting selection before and after propofol bolus infusion individually, according to the electrocorticography analysis standards from The Cleveland Clinic Foundation [14].

### 2.2. Anesthesia

After patients entered the operating room, invasive blood pressure, heart rate, electrocardiogram, respiratory motility, oxygen saturation, minimum alveolar concentration (MAC), bispectral index (BIS), etc. were monitored on a continuous basis. Propofol (AstraZeneca, Shanghai, China) 1.5–2.5 mg/kg, sufentanil (Yichang Renfu Pharmaceutical Co., Ltd., Yichang, Hubei, China) 0.1–5.0 μg/kg, and vecuronium (Nanjing Haijing Pharmaceutical Co., Ltd., Nanjing, China) 0.1 mg/kg were used to induce anesthesia. Intravenous propofol was tapered for 5 to 10 min prior to ECoG recording, and the other anesthetics were unchanged. At the same time, the MAC and BIS (fixed modules on the anesthesia machine) were controlled between 0.3~0.4 and 40~60, respectively, in order to minimize changes due to anesthesia [15]. The ECoG in and around the lesion was monitored following the injection of propofol (0.01 mg/kg). In addition, we compared IEDs prior to tumor resection versus after tumor resection.

### 2.3. ECoG Recording and Assessing

The ECoG was recorded using a 16-lead silica gel strip electrode (Huake Precision Medical Technology Co., Ltd., Beijing, China). Furthermore, a more stable period before the burst suppression occurred was selected to avoid confounding ECoG interpretation caused by burst suppression. The ECoG was recorded as an analogue signal using the Epoch XP Neurological Intraoperative Monitoring Workstation (Axon System, Hauppauge, NY, USA). Visual analysis of the ECoG, identification and labeling of IEDs, and parameter selection: low-cut filter 0.16 Hz, high-cut filter 30 Hz, sensitivity 200–500 μV/mm, and paper speed 30 mm/s. ECoG monitoring was performed by experienced neurosurgeons, anesthesiologists, and electrophysiologists. One of the authors who analyzed ECoG records had over 20 years of experience in clinical electroneurophysiology.

As shown in Figure 2, the ECoG recorded during period A (about 2 min), after the propofol bolus but before lesion resection, was considered the baseline, which was beneficial for subsequent quantitative and qualitative analyses of IEDs. After lesion resection, propofol was injected intravenously, and the ECoG at the same position was recorded. The following 2 min after propofol injection was named period B. Quantitative and qualitative comparisons were taken to analyze ECoG changes between periods A and B. If burst suppression occurred during ECoG recordings, only the ECoG before the emergence of burst suppression would be analyzed.

The activated ECoG was defined as follows:

After the infusion of low dose propofol, newly emerging IEDs were found in the contacts with no IEDs before the infusion of low-dose propofol (Figure 3A). After infusion of low-dose propofol, the amplitudes of IEDs in the contacts were increased, compared to the amplitudes before the infusion of low-dose propofol (Figure 3B). After the infusion of low-dose propofol, the frequency of IEDs increased in the contacts, compared to the frequency before the infusion of low-dose propofol (Figure 3C).

Meanwhile, in the epilepsy group, the activated group was subdivided into two groups, the treated and untreated groups, depending on whether or not the activated brain tissue accepted surgical treatment. In the treated group, the activated brain tissue was treated through cortical resection or repetitive thermos-coagulation until visual IEDs disappeared after intravenous propofol injection. In the untreated group, the propofol-activated brain tissue was not treated by surgery for various reasons, for example, IEDs were located in functional areas or the proximity of blood vessels.

### 2.4. Follow-Up

All patients in this study experienced at least a period of 12 months of follow-up. After discharge from the hospital, patients were usually reviewed at fixed clinical appointments. The early and long-term outcomes after surgery were assessed, respectively. Postoperative EPS refers to seizures occurring from 24 h to 2 weeks after surgery. The review content during follow-up included the 24 h ambulatory EEG monitoring (AEEG) and the Engel classification. All patients with epilepsy in the study experienced antiepileptic drug therapy according to the latest guidelines [5].

In the epilepsy group, all patients were given levetiracetam (Shenzhen Xinlitai Pharmaceutical Co., Ltd., Guangzhou Shenzhen, China) 500 mg bid (≥2 days) monotherapy, and levetiracetam (500 mg bid) was continued after surgery to prevent seizures. If the patients were postoperatively seizure-free, the drug could be discontinued 12 to 24 months after the seizures stopped.

The control group was not treated with any antiepileptic drugs before the operation. Given the increased risk of early postoperative seizures (EPS) due to surgery, levetiracetam (500 mg bid) was considered for 2 weeks after surgical resection in the control group.

If the epilepsy group or the control group had postoperative seizures, individualized treatment (adjusting the dosage, time, drug type, etc.) was given according to each patient’s condition.

The research object was selected for the non-rapid eye movement sleep (NREM) I-II period in the AEEG. The results of AEEGs were divided into four levels [16]: (1) non-sharp (NS) spike, (2) rare sharp (RS) spike (<4 in a 20 min sleep record), (3) moderate sharp (MS) spike (>4 and <1 per 10 s), and (4) large sharp (LS) spike (>1 per 10 s). If the EEG onset was captured during AEEG monitoring, the patient would be classified as the LS type. The NS and RS types could be considered favorable results, and the corresponding MS and LS were considered poor AEEG results. The results of the patient’s AEEG after 12 months were compared with those before surgery.

Engel classification [17]: Engel Class Ⅰ was considered a favorable seizure outcome, while Engel Classes Ⅱ-Ⅳ were considered unfavorable. Class I: The seizure disappeared completely or had only auras; Class II: only rare attacks, less than 3 per year; Engel III: more than 3 episodes per year, but the total decrease was 75 percent or more; Engel IV: seizures reduced by less than 75 percent.

### 2.5. Statistical Analysis

The Statistical Package for the Social Sciences version 27.0 (SPSS, Chicago, IL, USA) was used for data analysis and statistical processing in this study. Count data were assessed using Chi-squared (χ^2^) and Fisher’s exact tests. The probability (*p*) value and 95% confidence interval were applied in the study, and *p* < 0.05 was considered statistically significant.

## 3. Results

### 3.1. Effects of Low-Dose Propofol on ECoGs

A total of 408 patients were included in the study, and 102 patients were excluded because of lost follow-up. As a result, a total of 306 consecutive patients were enrolled; among them, 239 patients were included in the epilepsy group, and 67 patients were included in the control group. The demographic characteristics are shown in Table 1. No significant differences were found between the epilepsy and control groups in terms of age, sex, follow-up time, side, WHO grade, seizure phenotype, and so on.

According to the definition of activated ECoGs, in the epilepsy group, 177 patients who had activated ECoGs were included in the activated group, and 62 patients who had no activated ECoGs were included in the inactivated group. Meanwhile, in the control group, only six patients had activated ECoGs and were included in the activated group, and 61 patients had no activated ECoGs and were included in the inactivated group. Therefore, the activation rate of the epilepsy group was 74%, which was significantly higher than that of the control group (9%) (*p* < 0.001, Table 2).

Furthermore, in the activated group, 148 patients experienced surgical treatment on the activated brain tissue, according to the ECoG monitoring during surgery, and were included in the treated group. The other 29 patients included in the untreated group did not experience surgical treatment on the activated brain tissue because the locations were in functional areas or in proximity to blood vessels. All the patients in the control group did not experience surgical treatment on activated brain tissue because the patients had no epileptic seizures before surgery. For patients in the treated group, cortical resection and/or thermos-coagulation were performed on the activated brain tissue until visual IEDs disappeared after intravenous injection of low-dose propofol.

### 3.2. Outcomes of Epilepsy Patients after Surgery

#### 3.2.1. Early Outcomes of Epilepsy Patients

As shown in Table 3, during the long-term follow-up (24 h to 2 weeks after surgery), in the epileptic group, only 9 of 148 patients (6%) had EPSs, and 139 of 148 (94%) patients did not have EPSs. In the group of 29 patients with the untreated epilepsy, 6 of 29 patients (21%) had EPSs at least once, and 23 of 29 patients (79%) had no EPSs. The epilepsy patients of the untreated group had a higher occurrence of EPSs, compared with epilepsy patients in the treated group (*p* < 0.01).

#### 3.2.2. Long-Term Outcomes of Epilepsy Patients

During the long-term follow-up (12 months after surgery), 142 of 148 patients in the treated group had favorable seizure outcomes (Engel class I). In the untreated group, 22 of 29 patients had favorable seizure outcomes. In the treated group, 96% of patients had favorable seizure outcome rates, which was significantly higher than the 76% in the untreated group (Table 4, *p* < 0.001). At the same time, only 6 of 148 patients (4%) in the treated group had unfavorable outcomes (Engel Class II–IV), compared with 7 of 29 (24%) patients in the untreated group.

In addition, according to long-term follow-up AEEG results, 112 (76%) of the 148 patients in the treated group obtained good AEEG results (NS or RS), while 8 (28%) of the 21 patients in the untreated group obtained good AEEG results. The proportion of good AEEG results in the treated group was 76%, higher than 28% in the untreated group (Table 5, *p* < 0.001).

At long-term follow-up, some patients in both the treated and untreated groups had unfavorable AEEG results (MS or LS). Of the 148 patients in the treated group, there were 32 patients who had unfavorable AEEG results, representing 24% of the total number of patients in the treated group. Of the 29 patients in the untreated group, 13 patients had follow-ups for MS or LS, which represented 72% of the total number of the untreated group.

In the control group, only two patients had EPSs, and the frequency of seizures gradually diminished after administering levetiracetam (500 mg bid). During long-term follow-up, all patients achieved Engle Ⅰ, and the postoperative AEEG showed no obvious IEDs.

## 4. Discussion

The main result of this study was that the injection of low-dose propofol could activate the ECoG during GRE surgery, which could assist neurosurgeons in locating potential epileptic foci during the resection of glioma and epileptic foci.

### 4.1. The Effects of Propofol on ECoGs

Prior to the low-dose propofol infusion, the propofol used to maintain anesthesia was discontinued for 5 to 10 min, and other anesthetics were unchanged. Thus, in the present study, the transient changes in blood concentration caused by the low-dose infusion of propofol induced the changes in ECoG.

#### 4.1.1. Activated ECoGs Caused by the Infusion of Low-Dose Propofol

The present study showed that 177 of 239 (74%) patients in the epilepsy group had activated ECoGs caused by low-dose propofol injection (Figure 2). However, among patients without epilepsy, the rate of activated ECoGs was only 9%, indicating that a low dose of propofol could specifically activate ECoGs in patients with epilepsy. A previous study showed that the IEDs induced by propofol were mostly confined to the epileptogenic cortex in patients with epilepsy but hardly occurred in patients without epilepsy, which was consistent with our results in the present study [18]. Although there was no sufficient evidence to prove the potential epileptogenicity of propofol, many studies have reported that propofol could trigger epileptic seizures [12,13].

#### 4.1.2. The Change in ECoG Background Caused by the Infusion of Low-Dose Propofol

As shown in Figure 4, after a low-dose propofol injection, a period of background depression of the ECoG was observed in patients of both the epilepsy and control groups. At the same time, the frequency or amplitude of the ECoG in the activated brain tissue was not depressed or even increased, which helped us identify IEDs from the cluttered EEG background during the operation.

Zijlmans reported that propofol inhibited non-epileptic discharges [10]. Maschio also reported that propofol did not enhance non-epileptic discharges [4]. Therefore, a small dose of propofol might inhibit the conduction of non-epileptic discharges.

Therefore, these results suggested that low-dose propofol provided assistance in locating the epileptogenic zone and provided guidance on the effective surgical treatment of the epileptogenic zone during surgery, which might improve the control over the risk of postoperative epilepsy.

### 4.2. Guidance of Activated ECoGs during the Resection of Glioma and the Epileptogenic Zone

#### 4.2.1. The Effects of Low-Dose Propofol on the ECoG

Although the application of intraoperative ECoGs significantly improves the positive rate of locating epileptogenic zones, under the condition of general anesthesia, the use of anesthetic drugs may influence brain electrical activity; therefore, some epileptic foci and potential epileptiform discharge foci may be concealed, which may lead to a relapse of postoperative epilepsy after surgery [19,20]. A number of anesthetic drugs are known to influence the intra-operative ECoG [21], which may confound ECoG interpretation and the decision-making of surgical resection during epilepsy surgery [15].

For example, propofol’s effect on EEGs has been reported in numerous studies, but there is no consensus. Some studies suggest that propofol can activate EEGs, and some studies suggest that propofol can inhibit EEGs or have no significant effect on EEGs.

In this study, the infusion of low-dose propofol resulted in ECoG activation in GRE patients. In some patients with epilepsy, the frequencies and/or amplitudes of IEDs were higher after the infusion of low-dose propofol, whereas others had newly onset IEDs following low-dose propofol infusion. These results were in agreement with Hodkinson’s view, who first noted the activation of the ECoG by propofol [21], which suggested that propofol-activated brain tissue might be potential epileptic foci.

What is the meaning of the activated ECoG by low-dose propofol infusion, and should the activated brain tissue be processed?

#### 4.2.2. Surgical Treatment of Activated ECoGs Is Beneficial to Epilepsy Freedom

As shown in Table 3, in the treated group of the epilepsy group, only 6% of patients had postoperative seizures at the early stage (from 24 h to 2 weeks after surgery), and 96% of patients had better seizure control (Engle I) at the long-term (12 months postoperatively) follow-up. Meanwhile, in the untreated group of the epilepsy group, 21% of patients had at least one EPS at the early postoperative stage, and 76% of patients had better seizure freedom (Engle I) at the long-term follow-up.

Due to these findings, surgical treatment on activated brain tissue resulted in a greater chance of seizure freedom in GRE patients, both at the early stage and long term, after epilepsy surgery, suggesting that the activated ECoG provides a reliable guide in GRE surgical intervention.

#### 4.2.3. Surgical Treatment of Activated ECoGs Is Beneficial to AEEG Outcomes

The number of IEDs is closely related to seizures, and scholars generally think that the more IEDs, the higher the seizure frequency [22,23]. In this study, at the long-term follow-up, 76% of patients in the treated group had fewer IEDs (NS or RS) in AEEGs, while the number was 28% in the untreated group, indicating more patients in the treated group had favorable AEEG outcomes (NS or RS) (Table 5, *p* < 0.001). Therefore, the AEEG outcomes were similar to seizure freedom outcomes in the treated and untreated groups.

### 4.3. Propofol Infusion-Activated ECoGs Are Dose-Dependent

Propofol is widely used in general anesthesia, anti-epileptic therapy, and sedation in the intensive care unit. Usually, the mean dose of propofol to cause burst suppression is 88.2 mg [24]. As an antiepileptic drug, the mean loading dose of propofol is 2.0 mg/kg, and the maintenance dose is 1–10 mg/kg/h.

This study found that a relatively high dose (2.0 mg/kg) of propofol caused burst suppression in patients with or without epilepsy. Nevertheless, at a low dose (0.01 mg/kg, which was lower than normal anesthetic doses), propofol could specifically activate the ECoG in epilepsy patients.

Hisada reported that the de-inhibition of low-dose propofol makes the cerebral cortex hyperexcitable [13]. It is possible that this phenomenon is related to the alternative effect of propofol on ionic currents and channels in cortical neurons [18,25,26,27]. There are research findings where GABA was higher in tumors, compared with the peritumoral or normal cortices [28]. Propofol has also been shown to selectively modulate postsynaptic GABA-A receptors in the cortex [29,30]. Lenjten reported that propofol could induce the NREMI-II phase, and IED generation is often related to the NREMI-II phase [31]. In addition, propofol selectively suppressed sodium currents and calcium conductance, thus influencing the neuronal excitability of cortical neurons [11,25]. However, the exact mechanism is still not clear.

## 5. Limitations

While the study demonstrates the effect of propofol on the ECoG, we must acknowledge some potential study limitations. This was a single center study. At the same time, the identification of IEDs in ECoGs relied primarily on artificial visual analysis and was subjectively affected by electrophysiology physicians. For this reason, automated, multi-center research is required in the future. Future studies also need to explore the possible mechanism of low-dose ECoG activation by propofol in GRE patients.

## 6. Conclusions

The intraoperative ECoG is a reliable real-time method to demonstrate brain electrical activity and locate the epileptic foci during GRE surgery. A low-dose infusion of propofol can specifically activate ECoGs in epilepsy patients. Surgical treatment on the propofol-activated brain tissue results in fewer IEDs in post-operation AEEGs and a higher chance of seizure freedom in the early and long-term stages after epilepsy surgery. Thus, activated ECoGs induced by the low-dose propofol may provide reliable and real-time guidance in locating the actual and hidden epileptogenic zones.

## Figures and Tables

**Figure 1 brainsci-13-00597-f001:**
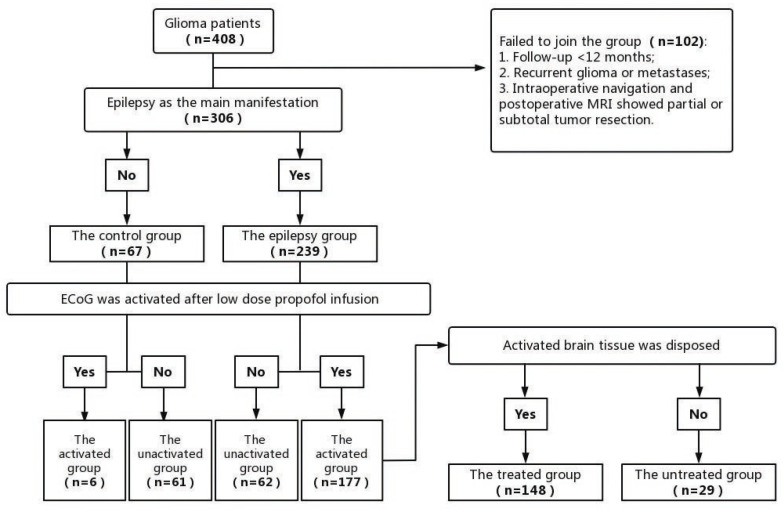
Patient enrollment process.

**Figure 2 brainsci-13-00597-f002:**
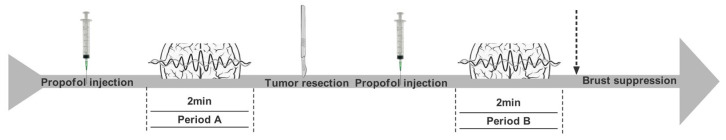
ECoG recording and assessing; Period A (about 2 min) after the propofol bolus (0.01 mg/kg) but before lesion resection; Period B (about 2 min) after lesion resection and the propofol bolus (0.01 mg/kg).

**Figure 3 brainsci-13-00597-f003:**
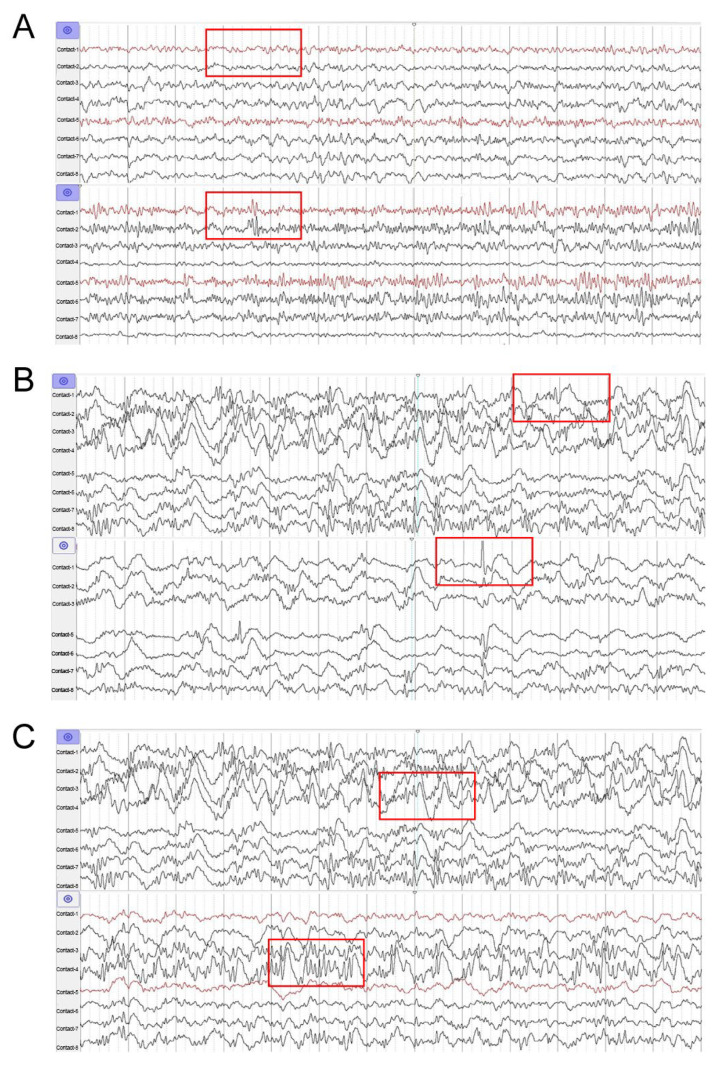
Three forms of ECoG activation by infusion of low-dose propofol. (**A**) Contact-2, the newly emerging IEDs after low-dose propofol injection. (**B**) Contact-1, the increased amplitude of IEDs after low-dose propofol injection. (**C**) Contact-4, the increased frequency of IEDs after low-dose propofol injection. (The most obvious changes in ECoG were marked with red boxes).

**Figure 4 brainsci-13-00597-f004:**
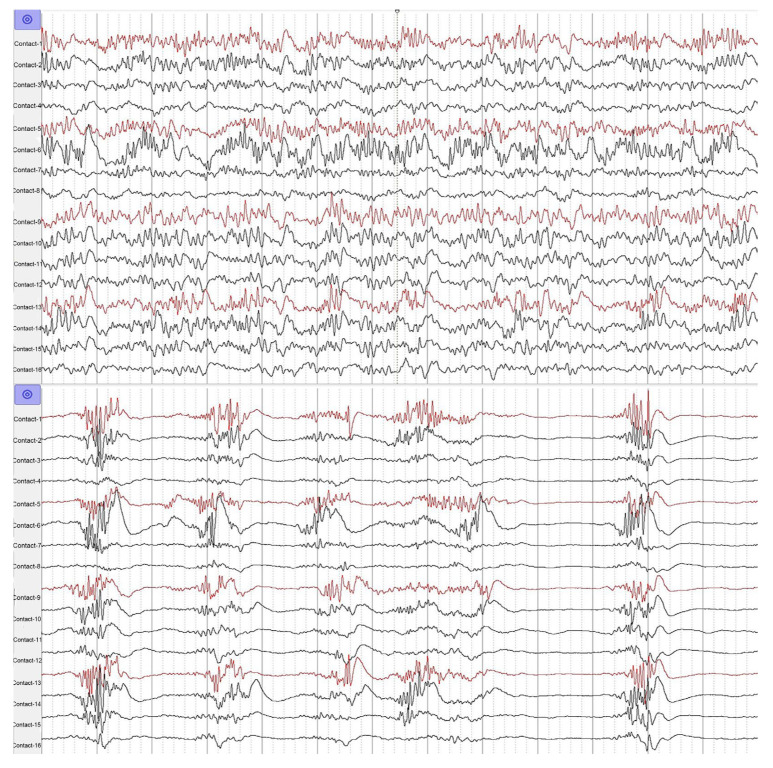
The low-dose propofol suppressed the background of ECoG.

**Table 1 brainsci-13-00597-t001:** The demographic characteristics of the epilepsy group and control group.

Characteristics	The Epilepsy Group	The Control Group	*p*-Value
No. of patients	239	67	
Preoperative AED therapy	Yes	No	
Mean age (years)	40.4	39.7	0.459
Males	144	42	0.612
Follow-up time (month)	18.5	19.3	0.528
Tumor side			0.376
Lt	102	25	
Rt	137	42	
Histological tumor subtype			0.652
Astrocytoma	72	19	
Mixed oligoastrocytoma	33	9	
Oligodendroglioma	95	27	
Ganglioglioma	27	8	
Others	12	4	
Seizure-onset features			0.668
Complex partial	13	4	
Generalized tonic-clonic	225	63	
Others	1	0	
Tumor location (lobe)			0.614
Temporal	66	19	
Frontal	123	35	
Parietal	36	10	
Occipital	14	3	
WHO grade			0.677
I	26	7	
II	122	34	
III	68	20	
IV	33	6	

**Table 2 brainsci-13-00597-t002:** The activation of ECoGs in the epilepsy group and control group.

	ECoG Activated	ECoG Not Activated	Incidence Rate (%)	*p*-Value
Epilepsy group (*n* = 239)	177	62	74	<0.001
Control Group (*n* = 67)	6	61	9	

**Table 3 brainsci-13-00597-t003:** Incidence of early postoperative seizures in the treated and untreated groups.

	With Early Seizure	Without Early Seizure	Incidence Rate (%)	*p*-Value
Treated Group (*n* = 148)	9	139	6	<0.01
Untreated Group (*n* = 29)	6	23	21	

**Table 4 brainsci-13-00597-t004:** The Engle class of epilepsy patients after surgery at the long-term follow-up.

	Engel Ⅰ	Engel Ⅱ–Ⅳ	Rate (%)	*p*-Value
Treated Group (*n* = 148)	142	6	96	<0.01
Untreated Group (*n* = 29)	22	7	76	

**Table 5 brainsci-13-00597-t005:** The AEEG outcomes of epilepsy patients after surgery at the long-term follow-up.

	NS and RS	MS and LS	Rate (%)	*p*-Value
Treated Group (*n* = 148)	112	36	76	<0.01
Untreated Group (*n* = 29)	8	21	28	

## Data Availability

The datasets generated and analyzed during the current study are not publicly available due to the fact that the files contain patient information; they can only be opened with certain software but are available from the corresponding author upon reasonable request.

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
