# Peer review of "Effects of Propofol on Cortical Electroencephalograms in the Operation of Glioma-Related Epilepsy"

_brainsci, 2023, doi:10.3390/brainsci13040597_

Round 1

Reviewer 1 Report

Comments and Suggestions for Authors

In this paper you barely touch on the matter of antiepileptic drug therapy. Did some of your patients receive such therapy prior to surgery, and perhaps went to surgery with residual conventional antiepileptic drugs still present in their brains? What was the nature of such prior therapy, if there was any? What was the nature of any post-surgery antiepileptic drug treatment and of its management? Would such treatment alter the interpretation of your findings e.g drug interactions etc?

You seem to use the words ‘treated’ and ‘treatment’ only in relation to surgical or other intra-operative treatment but I think readers may easily assume that you are writing about antiepileptic drug therapy. Could you distinguish clearly by appropriate wording, whereever necessary, between the two situations?

As I understand it, all of your patients had received substantial amounts of propofol before the cortical electrodes were put in place. Although that drug is often loosely described as having a very short half-life, this really refers to an early phase distributional half life after intravenous injection, and the terminal elimination half life is substantially longer (see, for instance, Sahinovic et al.(2018) Clinical Pharmacokinetics 57:1539-1558). I suspect your activating infusion of propofol was a bolus one (you mentioned the word bolus once) and that it was the falling phase of the quickly raised circulating level that activated epileptogenic mechanisms, just as abrupt withdrawal of another antiepileptic agent, a short half-life benzodiazepine that also acts on GABA mechanisms can do. Was the ECoG consistently assessed a fixed time after the (?) bolus dose?

In line 46 of the paper you  used the word ‘profound’ and I wonder if the word ‘potent’ might be more appropriate. Also, in relation to the electrocorticogram you write of ‘waves’ but at least some the appearances look more like epileptic spikes or sharp transients than actual waves.

Author Response

Please also see the attachment (the picture is in the attachment). 

Question 1: In this paper, you barely touch on the matter of antiepileptic drug therapy. Did some of your patients receive such therapy prior to surgery, and perhaps went to surgery with residual conventional antiepileptic drugs still present in their brains? What was the nature of such prior therapy, if there was any? What was the nature of any post-surgery antiepileptic drug treatment and of its management? Would such treatment alter the interpretation of your findings e.g drug interactions etc?

Response: Thank you for the above suggestion. We add the following in line 137 of the original text: According to guidelines, in the epilepsy group, after exclusion of contraindications (abnormal liver/renal function, history of acute/chronic hepatitis, and family history), levetiracetam (500 mg bid) was routinely administered in the perioperative period to prevent epileptic seizures, and the previous medication was continued after surgical resection. If postoperative seizures are free, the drug can be discontinued 12 to 24 months after the seizures stop. Given the increased risk of Early postoperative seizures (EPS) due to surgery, levetiracetam (500 mg bid) be considered for 2 weeks after surgical resection in the control group. (Liang et al., 2019)

Meanwhile, the control group and the epilepsy group were treated with antiepileptic drugs before and after surgery according to the guidelines. We think there is no significant effect on the results of this study.

Question 2: You seem to use the words ‘treated’ and ‘treatment’ only in relation to surgical or other intra-operative treatment but I think readers may easily assume that you are writing about antiepileptic drug therapy. Could you distinguish clearly by appropriate wording, whereever necessary, between the two situations?

Response: Thank you for the reviewer's advice to make my article easier to understand. Based on your advice, I have chosen a more rigorous vocabulary for the full text ' treated '. The surgical operation is collectively referred to as ' surgical treatment ' or ' treated by surgery ', while drug treatment is described as ' antiepileptic drug therapy '.

Question 3: As I understand it, all of your patients had received substantial amounts of propofol before the cortical electrodes were put in place. Although that drug is often loosely described as having a very short half-life, this really refers to an early phase distributional half life after intravenous injection, and the terminal elimination half life is substantially longer (see, for instance, Sahinovic et al.(2018) Clinical Pharmacokinetics 57:1539-1558). I suspect your activating infusion of propofol was a bolus one (you mentioned the word bolus once) and that it was the falling phase of the quickly raised circulating level that activated epileptogenic mechanisms, just as abrupt withdrawal of another antiepileptic agent, a short half-life benzodiazepine that also acts on GABA mechanisms can do. Was the ECoG consistently assessed a fixed time after the (?) bolus dose?

Response: In this study, we performed long-term EEG monitoring after tumor resection. However, only the EEG ( Period B ) of the first two minutes after propofol injection was selected for analysis. Because we have observed that the transient increase of propofol blood concentration activates the EEG, which is easily masked or inhibited by the subsequent burst inhibition.

The fast distribution half-life ( T a ) of propofol was 1.33-4.6 min, the slow distribution half-life ( T b ) was 27-69.3 min, the half-life ( T c ) was 116-834 min, the mean residence time ( MRT ) was 102-174 min, and the elimination half-life was 335 min. Before the injection of propofol, we stopped the intravenous propofol used to maintain anaesthesia about 5-10 min, while other anaesthetic drugs remained unchanged and controlled the minimum alveolar concentration (MAC) and bispectral index (BIS) between 0.3-0.4 and 40-60 to minimise the effects of anaesthetic drugs. It can be considered that the EEG changes observed in this study were only caused by changes in the concentration of propofol.

Question 4: In line 46 of the paper, you  used the word ‘profound’, and I wonder if the word ‘potent’ might be more appropriate. Also, in relation to the electrocorticogram you write of ‘waves’ but at least some the appearances look more like epileptic spikes or sharp transients than actual waves.

Response: We agree with the reviewer 's suggestions. We have changed the word ' profound ' to ' potent ' in line 46 and replaced all the word ' waves ' in the text with ' IEDs ' (interictal epileptiform discharges).

We would like to thank you for your careful reading, helpful comments, and constructive suggestions, which have significantly improved the presentation of our manuscript. Heartfelt thanks to you

Reviewer 2 Report

Comments and Suggestions for Authors

Dear authors,

 it's a great focus on the subject but i would like to tell you that this review needs modifications before considered, please see the following list of points which need to be considered:

Question 1 The introduction is very crowded and contains a lot of information that makes it difficult to read. I would suggest that the authors increase its size to make it lighter and more understandable.

Question 2: In the results section, the authors include several comments trying to explain the results, even though such comments help to understand the outcome, the integration of these comments along with the material and method information in the results section makes it very complicated and redundant. I would suggest to the author to focus the commenting on the results only in the discussion section.

Question 3: In the discussion section, there are many long sentences that make the reader lost the track. I would suggest the author to plan an outline that would organize and order their comments on the results in a more simple and understandable way.

In conclusion it is a good work with a great patient number and good statistical analysis. In my humble opinion by following the above advice it should be perfect.

My best regards.

Author Response

Question 1: The introduction is very crowded and contains a lot of information that makes it difficult to read. I would suggest that the authors increase its size to make it lighter and more understandable.

Response: Thank you for the suggestions. As suggested, we have improved the structure and organisation of the sentences and paragraphs. We hope the article has become easier to read.

Question 2: In the results section, the authors include several comments trying to explain the results, I would suggest to the author to focus the commenting on the results only in the discussion section.

Response: Thanks again to the reviewer for their valuable suggestions on the article’s structure. We have deleted the redundant discussion part of the results.

Question 3: In the discussion section, there are many long sentences that make the reader lost the track. I would suggest the author to plan an outline that would organize and order their comments on the results in a more simple and understandable way. 

In conclusion it is a good work with a great patient number and good statistical analysis. In my humble opinion by following the above advice it should be perfect.

Response: We thank the reviewer for reading our paper carefully and giving positive comments. According to the suggestion, we have re-adjust the structure of the discussion section and added subtitles to make it easier for readers to read and understand.

Thank you for your suggestions. All your suggestions are very important. They have important guiding significance for my thesis writing and scientific research work!

Round 2

Reviewer 1 Report

Comments and Suggestions for Authors

To an extent, the authors have evaded my query about pre-existing antiepileptic drug therapy by stating that a particular set of therapeutic guidelines was followed. As far as I can make out, these guidelines apply mainly to the immediate peri-operative situation and it seems likely that some of the patients studied would have previously had seizures treated with various antiepileptic drugs. If so, the paper does not make clear whether these previous drugs were discontinued before operation and, if so, how long before, since residual drug may have been present in the patients’ brains at the time of surgery.

There is also the ambiguous statement that previous therapy was continued after surgery without it being clear whether the previous therapy may have been the (probable) levetiracetam used in accordance with the guidelines that the authors mention, or antiepileptic drugs used previously..

It seems quite possible that other workers may consider it useful to follow the authors’ practices in the situation they describe if the paper is published. Therefore I think the authors should really make clear the nature of any previous antiepileptic drug therapy, whether continued or discontinued for the surgery, and also the nature of the antiepileptic therapy in the post-operative weeks. Without this information, others may obtain different outcomes using the authors’ procedures.

One of the references to the guidelines the authors used, #5, is published in the journal ‘Cancer Medicine’ and not in the ‘Journal of Cancer Medicine’ which does not appear to exist.

In the text, from line 105 on, there is a degree of conflict between the wording and the Figure. The ECoG in section A appears to have been recorded two minutes after the bolus propofol dose, but in part B  the ECoG is said to be recorded over a two minute time period at some unspecified time after the completed tumour removal propofol  bolus. However, in their response to my query it is stated that this part B ECoG analysis occurred two minutes after the bolus. It seems likely to me that the latter is what was done but the text of the revised paper needs revision to make this clear.

Author Response

To an extent, the authors have evaded my query about pre-existing antiepileptic drug therapy by stating that a particular set of therapeutic guidelines was followed. As far as I can make out, these guidelines apply mainly to the immediate peri-operative situation and it seems likely that some of the patients studied would have previously had seizures treated with various antiepileptic drugs. If so, the paper does not make clear whether these previous drugs were discontinued before operation and, if so, how long before, since residual drug may have been present in the patients’ brains at the time of surgery.

There is also the ambiguous statement that previous therapy was continued after surgery without it being clear whether the previous therapy may have been the (probable) levetiracetam used in accordance with the guidelines that the authors mention, or antiepileptic drugs used previously..

It seems quite possible that other workers may consider it useful to follow the authors’ practices in the situation they describe if the paper is published. Therefore I think the authors should really make clear the nature of any previous antiepileptic drug therapy, whether continued or discontinued for the surgery, and also the nature of the antiepileptic therapy in the post-operative weeks. Without this information, others may obtain different outcomes using the authors’ procedures.

One of the references to the guidelines the authors used, #5, is published in the journal ‘Cancer Medicine’ and not in the ‘Journal of Cancer Medicine’ which does not appear to exist.

Response: Sorry for the negligence in my manuscript. We have updated 5 # corresponding magazine: Cancer Medicine.

The patients in the epilepsy group were treated with levetiracetam 500 mg bid for at least two days before the operation and continued to be treated with levetiracetam 500 mg bid after the operation.

Patients in the control group were not treated with any antiepileptic drugs before surgery, and levetiracetam 500 mg bid was given 2 weeks after surgery to prevent epilepsy.

If the prognosis of patients in the epilepsy group or the control group is poor (Engel Class II-IV), individualized treatment (adjusting the dosage, time, drug type, etc.) is given according to the patient's condition.

If the patients in the epilepsy group had been treated with other antiepileptic drugs except for levetiracetam before surgery, the dose was gradually reduced and discontinued and replaced with levetiracetam 500 mg bid ( at least two days ).

We supplemented the description of antiepileptic drug therapy in the 2.4.Follow-up ( 144-153 lines ) :

In the epilepsy group, all patients were given levetiracetam 500mg bid (≥2 days) monotherapy, and levetiracetam (500mg bid) was continued after surgery to prevent seizures. If postoperative seizures free, the drug can be discontinued 12 to 24 months after the seizures stop.

The control group was not treated with any antiepileptic drugs before the operation. Given the increased risk of early postoperative seizures (EPS) due to surgery, levetiracetam (500 mg bid) be considered for 2 weeks after surgical resection in the control group.

If the epilepsy group or the control group has postoperative seizures, individualized treatment ( adjusting the dosage, time, drug type, etc. ) is given according to the patient's condition.

Thank you for your advice to make my manuscript more rigorous.

In the text, from line 105 on, there is a degree of conflict between the wording and the Figure. The ECoG in section A appears to have been recorded two minutes after the bolus propofol dose, but in part B  the ECoG is said to be recorded over a two minute time period at some unspecified time after the completed tumour removal propofol  bolus. However, in their response to my query it is stated that this part B ECoG analysis occurred two minutes after the bolus. It seems likely to me that the latter is what was done but the text of the revised paper needs revision to make this clear.

Response: Thank you again for your opinion. We have updated the description of period B: After resection of the lesion, propofol ( 0.01 mg/kg) was injected intravenously, and ECoG at the same position was recorded. And the following 2 min after propofol injection was named period B.